# Electrochemical Enantiomer Recognition Based on sp^3^-to-sp^2^ Converted Regenerative Graphene/Diamond Electrode

**DOI:** 10.3390/nano8121050

**Published:** 2018-12-14

**Authors:** Jingyao Gao, Haoyang Zhang, Chen Ye, Qilong Yuan, Kuan W. A. Chee, Weitao Su, Aimin Yu, Jinhong Yu, Cheng-Te Lin, Dan Dai, Li Fu

**Affiliations:** 1Key Laboratory of Marine Materials and Related Technologies, Zhejiang Key Laboratory of Marine Materials and Protective Technologies, Ningbo Institute of Materials Technology and Engineering (NIMTE), Chinese Academy of Sciences, Ningbo 315201, China gaojingyao@nimte.ac.cn (J.G.); yechen@nimte.ac.cn (C.Y.); Qilong.Yuan@nottingham.edu.cn (Q.Y.); Kuan.Chee@nottingham.edu.cn (K.W.A.C.); yujinhong@nimte.ac.cn (J.Y.); linzhengde@nimte.ac.cn (C.-T.L.); 2College of Materials and Environmental Engineering, Hangzhou Dianzi University, Hangzhou 310018, China; zhanghaoyang@hdu.edu.cn (H.Z.); suweitao@hdu.edu.cn (W.S.); 3Department of Electrical and Electronic Engineering, Faculty of Science and Engineering, University of Nottingham, Ningbo 315100, China; 4Department of Chemistry and Biotechnology, Faculty of Science, Engineering and Technology, Swinburne University of Technology, Hawthorn, VIC 3122, Australia; aiminyu@swin.edu.au

**Keywords:** graphene, chemical vapor deposition, electronic materials, enantiomer recognition, phenylalanine

## Abstract

It is of great significance to distinguish enantiomers due to their different, even completely opposite biological, physiological and pharmacological activities compared to those with different stereochemistry. A sp^3^-to-sp^2^ converted highly stable and regenerative graphene/diamond electrode (G/D) was proposed as an enantiomer recognition platform after a simple β-cyclodextrin (β-CD) drop casting process. The proposed enantiomer recognition sensor has been successfully used for d and l-phenylalanine recognition. In addition, the G/D electrode can be simply regenerated by half-minute sonication due to the strong interfacial bonding between graphene and diamond. Therefore, the proposed G/D electrode showed significant potential as a reusable sensing platform for enantiomer recognition.

## 1. Introduction

In stereochemistry, the characteristic of a substance that cannot coincide with its image is called chirality. Chirality is ubiquitous in nature, which is the most basic characteristics of life processes. Most of the organic compounds that constitute the life body are chiral molecules, such as proteins, polysaccharides and nucleic acids. A number of metabolic and regulatory processes in biological systems are closely related to stereochemistry, so chiral molecules participate in a series of enzymatic reactions, host and guest effects as well as other biological phenomena. In addition, chiral drugs often have only one enantiomer that shows a therapeutic effect, and the other may be ineffective or even have toxic side effects.

Nowadays, considerable attention has been paid to the research of chiral recognition. Several methods of chiral recognition have been developed including spectrum based [1], chromatography based [2] and sensor based technologies [3]. Although spectrum based and chromatography based methods could combine enantiomer separation and determination together, the operation process is still sophisticated, and detection time is also quite long. Therefore, the development of sensor based chiral recognition methods, specifically electrochemical sensors, is of great practical significance due to their simplicity, fast response, low cost, on-line and real-time detection. Electrochemical chiral sensors are based on the differences in the effects of chiral selectors and enantiomers, and this difference can be identified by conversion of redox probe signals into electrical signals. However, surface modification of the electrode by a chiral selector or its composite is still a complex process. For example, glassy carbon electrodes (GCE) are reusable, but the pre-polishing process has great influence on the performance of the bare electrode. At the same time, the chiral selectors or molecular imprinted polymers often affects the conductivity and further amplifies the instability of the GCE [4]. Screen printed electrode (SPE) are low cost and disposable electrodes, but it is difficult to use for molecular recognition after the chiral selector modification because of the poor electrical conductivity of the printed ink. In the past decade, the growing interest in graphene chemistry provided many graphene based fascinating composites for electrochemical sensor fabrication due to their excellent electrical and morphological properties [5,6,7,8]. The preparation of reusable graphene electrodes is an important topic in electrochemical sensing.

l-phenylalanine is an essential amino acid that human beings are unable to produce, and must be supplemented from food sources. l-phenylalanine deficiency can affect tyrosine synthesis, resulting in the decrease of thyroxine level and its metabolic activity [9]. Its enantiomer, d-phenylalanine, is commonly recommended to promote muscle stiffness, help with walking and speaking, and is therefore a useful health care product for treatment of Parkinson’s disease [10].

In this work, we proposed a new strategy for electrochemical chiral sensor fabrication, which transforms a diamond surface into a graphene film via sp^3^-sp^2^ conversion. The fabricated graphene/diamond (G/D) was successfully applied to d- and l-phenylalanine recognition after a simple β-CD drop casting process. Due to the extremely strong interfacial forces between graphene and diamond, the proposed G/D electrode can be simply regenerated via a half-minute of ultrasonic cleaning treatment.

## 2. Materials and Experimental

All reagents were analytical grade and used without any further purification process. Phosphate buffer solutions (PBS) were made by mixing 0.1 M KH_2_PO_4_ with Na_2_HPO_4_. High pressure high temperature (HPHT) synthetic diamond substrate with size of 3.5 × 3.5 × 1 mm^3^ was purchased from Shenzhen Tiantian Xiangshang Diamond Co. Ltd. (Shenzhen, China).

For graphene-diamond (G/D) electrode preparation, a HPHT diamond substrate was placed on a Cu plate surface and annealed at 1100 °C with 8 sccm H_2_ flow (≈0.14 Torr). Residue Cu was etched by immersing the G/D into a mixed solution (CuSO_4_:HCl:H_2_O = 1 g:50 mL:50 mL). The G/D was then partially encapsulated by silicone resin and with the graphene surface left open. The electrode was connected to Cu foil via silver paint (Figure 1A). For β-CD surface medication, 10 μL of β-CD solution (1 mM) was drop casted on the G/D electrode surface and dried naturally (β-CD/G/D). 

All electrochemical tests were performed on a CHI760E Electrochemical Analyzer. A β-CD/G/D, Pt wire and 3 M Ag/AgCl electrode were used as a working electrode, counter electrode and reference electrode, respectively.

## 3. Results and Discussion

Figure 1 shows the detail of graphene film growth via sp^3^-to-sp^2^ conversion process. Under high temperature, carbon atoms on HPHT defects centers will continue to diffuse into melted copper. Due to the low solubility of carbon in copper [11], the saturated carbon atoms will gradually precipitate and rearrange within the interface between copper and HPHT to form a graphene film. When graphene film completely blocked the contact between HPHT and copper, the entire catalytic reaction ended. As a result, a HPHT substrate with graphene surface was obtained, which can be used as a highly stable and regenerative electrode after encapsulation (Figure 1B,C,D) [12].

The surface morphology change of HPHT can be observed via SEM characterization. As shown in Figure 2A,B, due to the continuous diffusion of carbon atoms into copper, the surface of HPHT was etched into many island structures. XPS was used to characterize the chemical state of carbon on the surface of HPHT and G/D. As shown in Figure 2C, the surface of HPHT was basically composed of sp^3^ bonds, except for a small number of defective oxygen functional groups. In contrast, the sp^3^ bond (C—C) on the surface of G/D had nearly decreased by half, while the sp^2^ bond (C=C) had formed clearly, indicating the sp^3^ bond had successfully converted to sp^2^. The quality of formed graphene was evaluated using Raman mapping. As shown in Figure 2D, the ratio of I_2D_/I_G_ of graphene in most areas of G/D was between 0.9–1.2, indicating that it contained only a few layers. A full Raman spectrum is shown in Figure 2E. Since the graphene formed on diamond was only a few layers, the diamond peak was still visible. The sheet resistance of formed G/D was around 400 Ω/Υ.

In this work, l and d-phenylalanine were chosen for evaluating the feasibility of a G/D electrode for electrochemical enantiomer recognition. β-CD was chosen as a chiral selector due to its excellent solubility and chiral recognition property [13]. Linear sweep voltammetry (LSV) was used for electrochemical enantiomers recognition. Figure 3 shows the LSV profiles of G/D and β-CD/G/D electrodes for l and d-phenylalanine recognition. It can be seen that 0.1 mM l, d-phenylalanine oxidized at the same potential on the G/D electrode without β-CD modification (Figure 3A). Although the oxidation current of l-phenylalanine was slightly higher than that of d-phenylalanine, it could not be used to identify the type of phenylalanine with an unknown concentration. In contrast, l-phenylalanine and d-phenylalanine were oxidized at different potentials on the β-CD/G/D electrode (Figure 3B), which confirms the G/D electrode can be successfully used for l, d-phenylalanine recognition after a simple β-CD drop casting process. More specifically, the oxidation of _l_-phenylalanine was observed at 0.576 V with an enhanced current response. This enhancement could be ascribed to the strong interaction between β-CD molecules and l-phenylalanine, which concentrated more _L_-phenylalanine on electrode surface. This phenomenon is in agreement with other works [14,15]. In contrast, a suppressed oxidation of _D_-phenylalanine was observed at 0.550 V.

The reproducibility of the G/D electrode was tested by five individually fabricated electrodes. After the β-CD modification, these five electrodes all exhibited different oxidation potentials for l,d-phenylalanine. More specifically, the oxidation potential of the d-phenylalanine was always lower than the potential of l-phenylalanine oxidation. Because of the strong interaction between graphene and HPHT, G/D electrodes can be reused after a half-minute ultrasonic cleaning process in water with performance remaining very stable. The G/D electrode can be reused more than twenty times without sensing degradation. Currently, fabrication of graphene-based electrodes is a very hot area for biosensor development due to the outstanding properties of graphene. In most cases, the graphene or graphene-based composites are formed in a dispersion form and used as a modifier for commercial electrode modification, such as glassy carbon electrode or screen printed electrodes. These biosensors cannot be reused because the cleaning process could remove the modifier as well. In our reports, the G/D electrode can be simply regenerated by a half-minute sonication due to the strong interfacial bonding between the graphene and diamond. Therefore, the G/D enantiomer recognition sensor proposed in this work has more practical potential compared to modifier based electrochemical enantiomer recognition sensors.

## 4. Conclusions

In conclusion, a highly stable and regenerative G/D electrode was formed via a carbon sp^3^-to-sp^2^ conversion process. After a simple β-CD drop casting, the G/D electrode can be used as an electrochemical platform for l and d-phenylalanine recognition. Due to the strong interaction between graphene and HPHT, the proposed G/D electrode can be regenerated using a simple sonication cleaning process. Therefore, the G/D electrode has more practical potential compared to modifier based electrochemical sensors.

## Figures and Tables

**Figure 1 nanomaterials-08-01050-f001:**
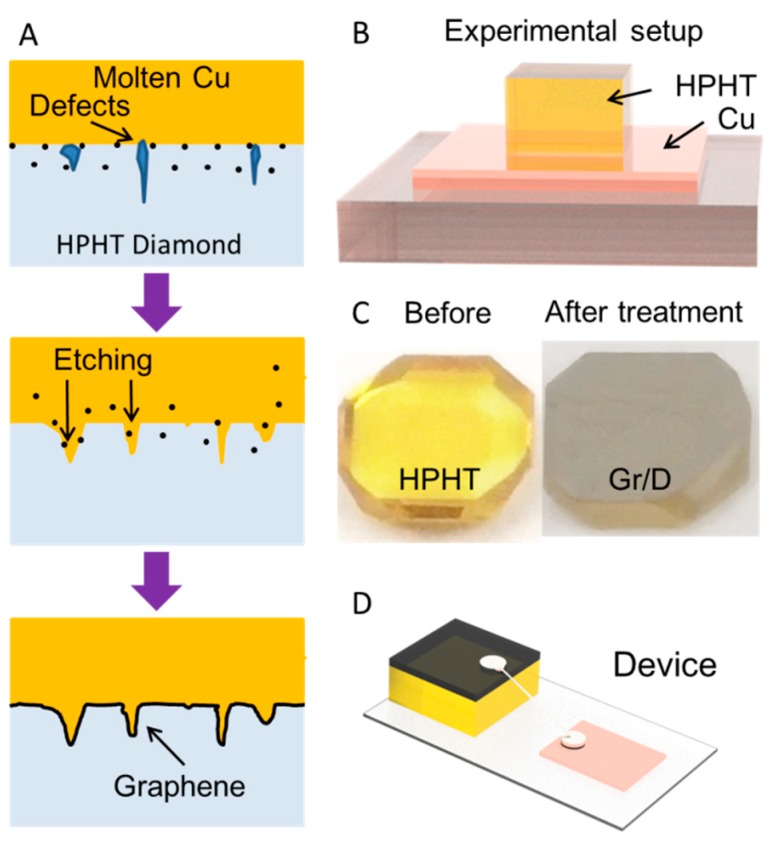
(**A**) Schematic diagram of graphene formation. (**B**) Scheme of preparation of graphene/diamond (G/D) electrode using chemical vapor deposition (CVD) method. (**C**) Photograph of sample before and after thermal treatment. (D) Scheme of the device after encapsulation

**Figure 2 nanomaterials-08-01050-f002:**
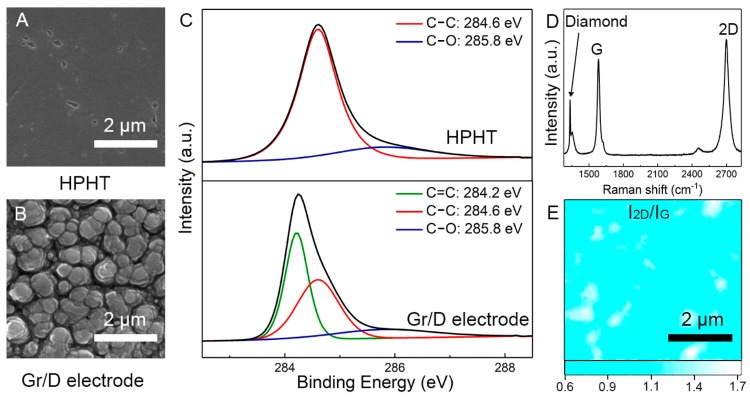
(**A**,**B**) SEM images and (**C**) XPS spectra of the high pressure high temperature (HPHT) and G/D electrode. (**D**) Raman mapping profile of I_2D_/I_G_ in a 5 × 5 μm area. (**E**) A typical Raman spectrum of G/D.

**Figure 3 nanomaterials-08-01050-f003:**
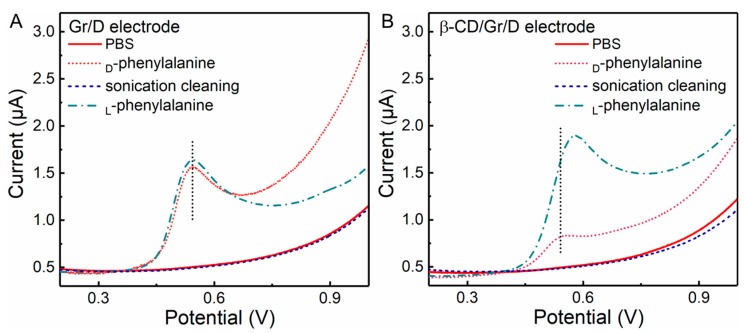
Linear sweep voltammetry (LSV) responses of (**A**) G/D and (**B**) β-CD/G/D in the absence and presence of 0.1 mM l- and d-phenylalanine in 0.1 M PBS (pH 7.0). Amplitude: 50 mV; Pulse width: 0.05 s; Pulse period: 0.5 s.

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
