# Peer review of "Electrochemical Enantiomer Recognition Based on sp3-to-sp2 Converted Regenerative Graphene/Diamond Electrode"

_nanomaterials, 2018, doi:10.3390/nano8121050_

Round 1

Reviewer 1 Report

The manuscript is about synthesis of regenerative sp3-to-sp2 converted graphene/diamond electrode and its possible application to recognition of enantiomers of phenylalanine. The manuscript is too short even to communication. Too short is in terms of presented results. The results might be interesting but it is unclear if they are reliable, since no reproducibility studies were performed.

English including needs some revision.

Title should be rewritten, since it is unusual in English start it from lower case.

Numbering of affiliations is inconsistent.

Information about material and analyte should be moved to introduction.

Reproducibility studies of the electrode material as well as electrode performance should be conducted additionally.

Figure captions are too short and do not include experimental parameters.

Conclusion is too general. 

Author Response

The manuscript is about synthesis of regenerative sp3-to-sp2 converted graphene/diamond electrode and its possible application to recognition of enantiomers of phenylalanine. The manuscript is too short even to communication. Too short is in terms of presented results. The results might be interesting but it is unclear if they are reliable, since no reproducibility studies were performed.

Response: Thank you very much for your valuable comments. We revised our manuscript based on your suggestion. We further conducted some experiments and included the results in the revised manuscript.

English including needs some revision.

Response: Thank you for your comments. We did a proofreading for our revised manuscript. All changes were highlighted.

Title should be rewritten, since it is unusual in English start it from lower case.

Response: Thank you for your suggestion. The title of our work has been changed to “Electrochemical enantiomer recognition based on sp3-to-sp2 converted regenerative graphene/diamond electrode”.

Numbering of affiliations is inconsistent.

Response: All affiliations were corrected. Thank you.

Information about material and analyte should be moved to introduction.

Response: Thank you for your suggestion. We moved these contents to Introduction section.

Reproducibility studies of the electrode material as well as electrode performance should be conducted additionally.

Response: We further did the reproducibility results in the manuscript. Please refer to the following content:

The reproducibility of the G/D electrode has been tested by five individual fabricated electrodes. After the β-CD modification, these five electrodes all exhibited different oxidation potentials for L, D-phenylalanine. More specifically, the oxidation potential of the D-phenylalanine always lowers than the potential of L-phenylalanine oxidation. For reusability test, The G/D electrode can be reused more than twenty times without sensing degradation.

Figure captions are too short and do not include experimental parameters.

Response: Thank you for your comment. We revised all figure captions. The details of all parameters were included.

Conclusion is too general.

Response: The conclusion has been re-written. Please refer to following content:

In conclusion, a highly stable and regenerative G/D electrode was formed via carbon sp3-to-sp2 conversion process. After a simple β-CD drop casting, the G/D electrode can be used as electrochemical platform for L and D-phenylalanine recognition. Due to the strong interaction between graphene and HPHT, the proposed G/D electrode can be re-generated using a simple sonication cleaning process. Therefore, the G/D electrode has more practical potential compared to the modifier based electrochemical sensor.

Reviewer 2 Report

row 98-117 or parts must be in introduction, motivation

row 69 Figure 1A - A not described in fig.

10 µl  liter with low l SI units

row 90 (C-C) not in linebreak

Fig 2C and D the axes description the letters are to small

Fig2D intensity scale is not described

In conclusion no description about potential to use this, this was described before in row 98 - 117

Author Response

row 98-117 or parts must be in introduction, motivation

Response: Thank you for your suggestion. We moved these contents to Introduction section.

row 69 Figure 1A - A not described in fig.

Response: We are sorry for making you confused. The Figure 1 has been modified.

10 µl  liter with low l SI units

Response: We corrected the unit.

row 90 (C-C) not in linebreak

Response: We corrected the symbol.

Fig 2C and D the axes description the letters are to small

Response: Thank you for your comment. We changed the font size of these letters.

Fig2D intensity scale is not described

Response: Thank you for your comment. The intensity scale of Fig 2D represented the ratio of ID/IG. The depth of the blue color indicates the value of ID/IG ratio change. We also added some description in the revised manuscript.

In conclusion no description about potential to use this, this was described before in row 98 – 117

Response: Thank you for your comment. The conclusion section has been re-written. The potential practical application has been included in the new version.

Reviewer 3 Report

The authors use a diamond-graphene electrode drop-casted with beta-cyclodextrin to recognize D and L phenylalanine. The introduction and some of the experimental results lack of clarity and reference to other literature work. The authors should address the following issues:

1.      Abstract. Please make more clear and fluent the sentence: ”In this work…. ”

2.      Materials and experimental. Please give references to the graphene/diamond fabrication process. What is the impact of different temperature, time and pressure on the graphene formation process?

3.      Line 93. Raman mapping, Figure 2D. Please add axes and label to the colorbar. Why do you say that the ID/IG ranges between 0.6 to 1.2 if your colorbar ranges 0.6-1.7 ?
Can you please show a typical Raman spectrum ? Is the diamond peak at 33300px-1 still visible ?

4.      Figure2B. Your material is very rough, is the graphene conformal to the surface or you have only isolated spots with graphene ?

5.      Line 105. Without the beta-CD your electrode is not able to recognize L and D PA, what is the then the benefit of using your electrode compared to others ?

6.      Line 118. How much do the electrode performances (voltage shift) degrade after 1 cycle ?

Author Response

1.      Abstract. Please make more clear and fluent the sentence: ”In this work…. ”

Response: Thank you for your suggestion. The Abstract section has been modified. Please refer to following content:

It is of great significance to distinguish enantiomers due to their different, even completely opposite biological, physiological and pharmacological activities. A sp3-to-sp2 converted highly stable and regenerative graphene/diamond electrode (G/D) was proposed as an enantiomer recognition platform after a simple β-cyclodextrin (β-CD) drop casting process. The proposed enantiomer recognition sensor has been successfully used for D and L-phenylalanine recognition. In addition, the G/D electrode can be simply regenerated by half minute sonication due to the strong interfacial bonding between graphene and diamond. Therefore, the proposed G/D electrode showed a significant potential as a reusable sensing platform for enantiomer recognition.

2.      Materials and experimental. Please give references to the graphene/diamond fabrication process. What is the impact of different temperature, time and pressure on the graphene formation process?

Response: Thank you for your comment, the reference has been added in the revised manuscript. Once the reaction temperature below the copper melting point( 1083.4 ℃),diamond unable to have sufficient contact with copper, result in none graphene formation on diamond. Raising the reaction temperature will accelerate the formation of graphene. However, it has few impacts on graphene quality. The reaction time have an impact on integrality of graphene on diamond, once the diamond surface is covered with graphene, prolonging the reaction time has few impacts on graphene quality. We have ever changed the reaction pressure, however, we have not found some impact on graphene formation process, perhaps we can make a detailed investigation of the effect of pressure in another work.

3.      Line 93. Raman mapping, Figure 2D. Please add axes and label to the colorbar. Why do you say that the I2D/IG ranges between 0.6 to 1.2 if your colorbar ranges 0.6-1.7 ?

Can you please show a typical Raman spectrum ? Is the diamond peak at 33300px-1 still visible?

Response: Thank you for raising this concern. Figure 2D has been modified according to your suggestion. The color bar of the Figure 2D indeed in the ranges from 0.6-1.7. However, the I2D/IG of the most area of the G/D is in the range between 0.6 to 1.2. To avoid misunderstanding, we modified the relative discussion in the revised manuscript.

The Raman spectrum is shown in Figure 2E, since the graphene formed on diamond is few layers, the diamond peak is still visible.

4.      Figure2B. Your material is very rough, is the graphene conformal to the surface or you have only isolated spots with graphene ?

Response: Thank you for raising this concern, based on the Raman mapping result (scan area: 5×5 μm), the graphene is formed on the entire surface of the diamond. In addition, we cannot receive any electrochemical signal if the graphene only formed on the isolated spots.

5.      Line 105. Without the beta-CD your electrode is not able to recognize L and D PA, what is the then the benefit of using your electrode compared to others ?

Response: Currently, fabrication of graphene-based electrode is a very hot area for biosensor development due to the outstanding property of the graphene. In most cases, the graphene or graphene-based composites were formed in a dispersion form and used as a modifier for commercial electrode modification, such as glassy carbon electrode or screen printed electrode. There biosensor cannot be reused because the cleaning process could remove the modifier as well. In our reports, the D/G electrode can be simply regenerated by half minute sonication due to the strong interfacial bonding between graphene and diamond. Therefore, the D/G enantiomer recognition sensor proposed in this work has more practical potential compared to the modifier based electrochemical sensor.

6.      Line 118. How much do the electrode performances (voltage shift) degrade after 1 cycle ?

Response: A slight positive voltage shift (about 0.03 V) with significant current degradation can be noticed after 1 cycle due to the surface fouling effect. Therefore, a reusable electrochemical enantiomer recognition sensor is essential for practical application.  

Round 2

Reviewer 1 Report

The manuscript is clear and can be published after correction of a digital mistake in line 178, where units while converting to pdf where corrupted.

Reviewer 2 Report

Fig. 2. C not readable the legend Fig. 2 E to small, nothing clear to read Generally the content has very low new aspects and interest to the reader.